# Effects of 3-Hydroxy-3-methylglutaryl-CoA Reductase Inhibitors on Cholesterol Metabolism in Laying Hens

**DOI:** 10.3390/ani13111868

**Published:** 2023-06-03

**Authors:** Huanbin Wang, Kuntan Wu, Xiaomei Mi, Shahid Ali Rajput, Desheng Qi

**Affiliations:** 1Department of Animal Nutrition and Feed Science, College of Animal Science and Technology, Huazhong Agricultural University, Wuhan 430070, China; whbin@webmail.hzau.edu.cn (H.W.); kuntanwu@webmail.hzau.edu.cn (K.W.); mixiaomei@webmail.hzau.edu.cn (X.M.); 2Faculty of Veterinary and Animal Science, Muhammad Nawaz Shareef University of Agriculture, Multan 60000, Pakistan; shahid.ali@mnsuam.edu.pk

**Keywords:** cholesterol metabolism, HMGCR inhibitor, laying hen, gene expression, RNA-seq

## Abstract

**Simple Summary:**

The cholesterol content in egg yolk is high, and an excessive consumption of egg yolk can have adverse effects on the health of the body. Therefore, it is necessary to systematically study the formation process of cholesterol in egg yolk and reduce the content of cholesterol in egg yolk. After using 3-hydroxy-3-methylglutaryl-CoA reductase (HMGCR) inhibitors to inhibit the synthesis of cholesterol in the liver of laying hens, this study investigated the changes in the cholesterol content in egg yolk and the surrounding tissues and utilised RNA sequencing (RNA-seq) technology to explore the biological mechanism that is involved when HMGCR inhibitors reduce egg yolk cholesterol. Therefore, this study provides a reference for related research on cholesterol metabolism in laying hens.

**Abstract:**

This study aimed to investigate the effect of HMGCR inhibitors on egg yolk cholesterol content and its biological mechanisms. Four groups of 180-day-old laying hens (*n* = 8 cages/group, 6 laying hens/cage) were fed a corn/soybean-based diet (control) and the control diet supplemented with an HMGCR inhibitor at 60, 150, and 300 mg/kg for 4 weeks. The experimental results showed that adding HMGCR inhibitors of 150 mg/kg or more can significantly reduce the cholesterol content in the liver, yolk, serum, and pectoral muscles of laying hens. The RNA-seq results showed that compared with the control group, the addition of HMGCR inhibitors of 150 mg/kg or more to the diet significantly upregulated genes related to cholesterol synthesis in the liver, and the genes involved in steroid synthesis and metabolism, sterol synthesis and metabolism, and cholesterol synthesis and metabolism were all affected by the HMGCR inhibitors. In summary, adding HMGCR inhibitors of 150 mg/kg or more to the diet of hens can significantly reduce the cholesterol content in egg yolk. After the HMGCR inhibitors inhibited the activity of the liver HMGCR, they also altered the expression of genes related to cholesterol synthesis, bile acid synthesis, and cholesterol transport in the liver, and ultimately reduced cholesterol synthesis and cholesterol transport to the egg yolk.

## 1. Introduction

Cholesterol in the diet is an important source of cholesterol in the human body, and excessive intake can cause imbalances in the body’s cholesterol homeostasis [1]. Due to the high cholesterol content in egg yolk (approximately 200 mg/egg) [2], in the mid-20th century, a large number of researchers suggested limiting the consumption of foods rich in cholesterol, such as eggs [3]. Meanwhile, other studies have proved that there is a close relationship between the consumption of eggs and the incidence rate of cardiovascular diseases caused by diabetes [4,5,6]. In addition, the American Heart Association (AHA) recommended in 2019 that patients with dyslipidaemia limit their intake of egg yolk cholesterol [7]. From this, it can be seen that the content of cholesterol in egg yolk has an impact on human health, so it is necessary to systematically study the formation process of cholesterol in egg yolk and reduce the content of cholesterol in egg yolk.

The laying hens mainly feed on plant-based feed, so the cholesterol required for their metabolic activities is almost entirely synthesised by the liver [8]. The cholesterol synthesised by the liver is transported to the bloodstream in the form of lipoprotein [9]. There are four kinds of lipoprotein in the blood that are involved in the transport of cholesterol: portomicrons, high-density lipoprotein (HDL), low-density lipoprotein (LDL), and very low-density lipoprotein (VLDL) [10]. The VLDL synthesised by laying hens is rich in vitellogenin and has a small volume, which is not easily degraded by lipoprotein lipase [11]. Only VLDL can transport cholesterol to the ovaries and participate in the formation of egg yolks [12]. 

HMGCR inhibitors are a class of statins that can specifically inhibit liver HMGCR activity [13,14]. Numerous studies have shown that HMGCR inhibitors can inhibit the synthesis of cholesterol in the human liver and thereby prevent hyperlipidaemia caused by high cholesterol levels [15,16,17]. Lui et al. reported that a daily intake of 20 mg of HMGCR inhibitors significantly reduced patients’ serum LDL-C levels [18]. De Haan et al. reported that adding 100 mg/kg of HMGCR inhibitors to the diet of mice significantly reduced total cholesterol levels in serum [19]. In addition, Elkin et al. reported that adding HMGCR inhibitors of 300–600 mg/kg to the diet of laying hens significantly reduced the egg weight and cholesterol content in egg yolk [20]. Meanwhile, Luhman et al. reported that adding 35 mg/kg of HMGCR inhibitors to the diet did not significantly affect the cholesterol content in egg yolk [21]. From this, it can be seen that there is a dose effect of HMGCR inhibitors on egg yolk cholesterol deposition, and the addition of high-dose HMGCR inhibitors has adverse effects on egg weight. However, there have been no reports on the effects of low-dose (60–300 mg/kg) HMGCR inhibitors on egg yolk cholesterol deposition and egg weight. 

In order to investigate the effect of low-dose HMGCR inhibitors on egg yolk cholesterol content and its biological mechanisms, atorvastatin calcium (60–300 mg/kg) was used as the HMGCR inhibitor in the diet in this study. The changes in cholesterol content in the liver, yolk, serum, and pectoral muscles of laying hens were studied, and RNA-seq and a real-time q-PCR were used to explore the biological mechanism of HMGCR inhibitors reducing egg yolk cholesterol. Therefore, this study provides a reference for related research on cholesterol metabolism in laying hens.

## 2. Materials and Methods

### 2.1. Laying Hens, Treatments, and Sample Collection

The animal protocol was approved by the Animal Ethics and Use Committee of Huazhong Agricultural University with approval number HZAUCH-2022-2012. A total of 192 180-day-old Jingfen No. 6 laying hens were randomly divided into 4 groups, with each group divided into 8 cages with 6 laying hens/cage. All the laying hens were allowed to freely access water and a corn/soybean-based diet (Appendix A), and the diet met the nutritional requirements for laying hens [22]. The temperature of the chicken coop was controlled between 20 and 22 °C, and the photoperiod was 16L:8D [23]. Atorvastatin calcium, which was purchased from McLean Biochemical Technology Co., Ltd. (Shanghai, China), was selected as the inhibitor of HMGCR (purity ≥99.0%) and was added to the diet of the test groups at the doses of 60, 150, and 300 mg/kg. The experiment lasted for 4 weeks, and the eggs were collected in the 2nd and 4th weeks; the egg yolks were separated, and blood was collected from the veins beneath the wings. The blood was centrifuged at a speed of 3000 rpm at 4 °C for 10 min to separate the bleeding serum [24]. The serum and egg yolk were stored at −80 °C until analysis. At the end of the experiment, the laying hens were euthanised through acute blood loss, and the liver, ovaries, and pectoral muscles were collected and stored at −80 °C for subsequent analysis. Meanwhile, a portion of the liver was collected and stored in a 4% formalin solution for later analysis of the liver pathology sections.

### 2.2. Determination of Egg Quality

After weighing the eggs, the yolks were separated and weighed. The ratio of egg yolk weight to egg weight was considered the egg yolk ratio. The Haugh unit, albumen height, and yolk colour were measured with a multifunctional egg quality tester (Beijing Brad Technology Development Co., Ltd., Beijing, China).

### 2.3. Determination of Cholesterol Content

The cholesterol content in the liver, egg yolk, serum, and pectoral muscles was measured by using a cholesterol reagent kit (Solebao Biotechnology Co., Ltd., Beijing, China). The levels of HDL-C, LDL-C and VLDL-C in the serum were measured by using corresponding commercial reagent kits (Nanjing Jiancheng Biotechnology Co., Ltd., Nanjing, China). The content of triglycerides (TG) in the serum was determined by using a triglyceride reagent kit (Solebao Biotechnology Co., Ltd., Nanjing, China). The product of the egg yolk cholesterol content and egg weight was the total cholesterol (TC) content of the egg. The ratio of the egg yolk cholesterol content to egg weight was the proportion of the egg cholesterol.

### 2.4. Liver Histology

Liver tissue blocks were fixed overnight in 4% formalin; then, they were embedded in paraffin blocks, sliced, and stained with haematoxylin and eosin (H&E).

### 2.5. Transcriptome Analysis

Total RNA isolation, mRNA purification, library preparation, and sequencing analysis were performed on the liver of the control group and two experimental groups with HMGCR inhibitor additions of 150 and 300 mg/kg (*n* = 6) [24]. The quality and quantity control results of the purified RNA are shown in Table 1, which could meet the requirements for machine sequencing. After sequencing, the two test groups to which the HMGCR inhibitor was added were compared with the control group, and the differential expression of genes (DEGs) was analysed by using the DESeq method. Genes with an expression fold change ≥2 and *p* < 0.05 were selected as significant DEGs. All significant DEGs were analysed via gene ontology (GO) enrichment and Kyoto Genes and Genomes (KEGG) enrichment.

### 2.6. Real-Time q-PCR Analysis

To evaluate the accuracy of the RNA-seq results, five genes, i.e., 3-hydroxy-3-methylglutaryl CoA reductase (HMGCR), mevalonate kinase (MVK), farnesyl diphosphate synthase (FDPS), farnesyl diphosphate farnesyltransferase 1 (FDFT1), and squalene epoxide enzyme (SQLE), were randomly selected for a real-time q-PCR analysis. In addition, the expression of the very low-density lipoprotein receptor (VLDLR) gene in the ovaries was also measured. The total RNA of the liver and ovary was extracted with a Trizol reagent (Solebao Biotechnology Co., Ltd., Beijing, China), and cDNA was synthesised via reverse transcription by using the ABScript III RT Master Mix (Vazyme, Nanjing, China). A real-time q-PCR was completed on a real-time fluorescent quantitative PCR instrument (Applied Biosystems, Waltham, MA, USA), and the experimental procedures and real-time q-PCR conditions were the same as those in the instructions for use. The 2^−ddCt^ method was used for the quantification with β-actin as a reference. All of the primers are listed in Appendix A.

### 2.7. Statistical Analysis

The experimental results were analysed by using IBM SPSS Statistics 25 software. First, all of the data were tested for normal distribution, and then the single factor analysis of variance was used to analyse the impact of HMGCR inhibitors on all of the tested indexes and q-PCR results. If there was a main effect, comparisons of multiple means were performed by using the Tukey–Kramer test.

## 3. Results

### 3.1. Effect of HMGCR Inhibitors on Egg Quality

The changes in egg quality during the second and fourth weeks are shown in Table 2. With the increase in the HMGCR inhibitor addition, the egg weight and yolk weight decreased significantly in the second and fourth weeks (*p* < 0.05), and there was no significant change in the Haugh unit and albumen height. In the fourth week, the high-dose HMGCR inhibitors reduced the egg yolk ratio and egg yolk colour (*p* < 0.05). 

### 3.2. The Effect of HMGCR Inhibitors on the Content of TC and TG in Tissues

The results of the changes in the TC and TG contents during the second and fourth week are shown in Table 3. In the second and fourth weeks, the content of serum cholesterol significantly decreased with the increase in the HMGCR inhibitor addition (*p* < 0.05). When the dose reached 300 mg/kg, the content of the lipoprotein cholesterol in the serum and the cholesterol content in the chest muscle tissue significantly decreased in the second and fourth weeks (*p* < 0.05). When the addition of the HMGCR inhibitor reached 150 mg/kg and above, the content of TG in the serum decreased significantly in the second and fourth weeks (*p* < 0.05).

### 3.3. The Effect of HMGCR Inhibitors on Cholesterol Content in Eggs

The changes in the cholesterol content in the eggs during the second and fourth weeks are shown in Table 4. When the addition of the HMGCR inhibitors reached 150 mg/kg and above, the egg yolk cholesterol content as well as the egg cholesterol percentage decreased significantly (*p* < 0.05) in the second and fourth weeks. The amount of total egg cholesterol decreased significantly with the increasing HMGCR inhibitor addition (*p* < 0.05).

### 3.4. Results of Liver Histology

The results of the liver H&E staining are shown in Figure 1A. The HMGCR inhibitors had no significant effect on the liver tissue structure of the laying hens. 

### 3.5. Sequencing, De Novo Assembly, and Annotation Analysis

A total of 18 libraries were obtained from three groups (control, 150, and 300 mg/kg HMGCR inhibitor test groups), in which the 150 bp-paired end of 45,215,482 to 62,788,508 raw reads were collected, and the Q30 values ranged from 93.55% to 94.56% (Table 1). After filtering out the low-quality reads, an average of 49,989,130 high-quality and clean reads were obtained. Then, high-quality and clean reads were further mapped onto the Gallus gallus genome (Ensembl database) by using the Bowtie and TopHat tools, and the average rate of the high-quality reads was 92.17% (Table 1).

### 3.6. Differential Expression and Functional Analysis of Genes

The results of the analysis of the DEGs are shown in Figure 1B,C. When the dosage of the HMGCR inhibitor was 150 mg/kg, a total of 309 genes were differentially expressed, of which 165 genes were upregulated and 144 genes were downregulated (Figure 1B; Appendix A). When the dosage of the HMGCR inhibitor was 300 mg/kg, a total of 803 genes were differentially expressed, of which 421 genes were upregulated and 382 genes were downregulated (Figure 1B; Appendix A). Through a coexpression analysis of DEGs in two groups with different dosages, it was found that 165 DEGs were coexpressed (Figure 1C). In addition, a GO enrichment analysis of the DEGs showed that the six GO categories with high enrichment related to cholesterol metabolism were the steroid metabolism process, steroid biosynthesis process, sterol metabolism process, sterol biosynthesis process, cholesterol metabolism process, and cholesterol biosynthesis process (Figure 2A). We also conducted a KEGG pathway enrichment analysis on the DEGs. When two different doses of HMGCR inhibitors were added, the pathways with the highest enrichment of DEGs were both involved in the steroid biosynthesis process (Figure 2B). The DEGs associated with hepatic cholesterol synthesis, bile acid synthesis, cholesterol transport, and reverse cholesterol transport in the GO and KEGG enrichment analyses are summarised and listed in Table 5. The real-time q-PCR results showed that compared with the control group, the addition of 150 and 300 mg/kg of HMGCR inhibitors significantly increased the mRNA expression levels of HMGCR, MVK, PDPS, FDFT1, and SQLE genes in the liver (*p* < 0.05) (Figure 2C), and the results obtained were similar to the RNA-seq results, which suggests that the results of the RNA-seq are reliable.

### 3.7. Changes in the Expression of the Ovarian VLDLR Gene

The addition of 60 mg/kg HMGCR inhibitor did not significantly affect the expression level of the ovarian VLDLR gene. When the dosage reached 150 mg/kg or above, the expression level of the VLDLR gene was significantly upregulated (*p* < 0.05) (Figure 2D).

## 4. Discussion

The liver uses acetyl CoA as the raw material and undergoes multistep catalytic reactions to generate HMG-CoA [1]. HMG-CoA is then catalysed by HMGCR to generate mevalonic acid (MVA), which ultimately forms cholesterol through multistep reactions [1]. Due to HMGCR being a rate-limiting enzyme in the cholesterol synthesis pathway, HMGCR inhibitors (statins) are commonly used to reduce liver cholesterol synthesis [13,14,15,16,17,18,19,20]. According to this study, HMGCR inhibitors have a dose effect on the cholesterol content in egg yolk, and 300 mg/kg is the minimum effective dose added to the diet. Meanwhile, the results of this study showed that low doses (60 mg/kg) of the HMGCR inhibitor had no significant effect on the egg yolk cholesterol content, whereas medium and high doses (150 mg/kg and above) of the HMGCR inhibitor significantly reduced the egg yolk cholesterol content. It can be seen that there was also a dose effect of the HMGCR inhibitors on the cholesterol content in the egg yolk, which is consistent with previous research results [20,21]. In addition, the minimum effective dose of the HMGCR inhibitor was found to be 150 mg/kg in this study, which was half of the previously reported dose.

The main effect of HMGCR inhibitors is the reduction in cholesterol in the serum [18,19]. In this study, adding HMGCR inhibitors of 150 mg/kg and above to feed can significantly reduce the content of TC and TG in the liver and serum of laying hens, which proves that HMGCR inhibitors also have good lipid-lowering effects on laying hens, which is consistent with the results reported by Elkin et al. [20].

The cholesterol in the egg yolk mainly comes from the liver, and adding HMGCR inhibitors of 150 mg/kg or more to feed can significantly reduce the cholesterol content in egg yolk; however, when the HMGCR inhibitor was added at 60 mg/kg, there was no significant reduction in the cholesterol content in the yolk. The reason may be that the serum level of the VLDL cholesterol (VLDL-C), which is responsible for transporting cholesterol to the ovary, was not significantly decreased when the HMGCR inhibitor was added at 60 mg/kg. It is also possible that this was due to an increase in the total amount of cholesterol synthesised by the ovary itself [11,20]. In addition, in this study, high-dose HMGCR inhibitors reduced the egg weight and yolk weight, which is consistent with the results reported by Elkin et al. [20]. The reason for this could be that the amount of VLDL synthesised by the liver also decreased after the amount of hepatic cholesterol synthesis was reduced, which affected the transport of lipids into the yolk. It is also possible that damage to the reproductive system of laying hens by HMGCR inhibitors causes a decrease in egg weight and yolk weight, and the specific reasons need to be further investigated. 

Studies of the transcriptome response of laying chicken liver to HMGCR inhibitors have not been reported. The amount of cholesterol in egg yolk decreased significantly when HMGCR inhibitors were added at doses of 150 mg/kg and above; therefore, control groups and two experimental groups with additives of 150 and 300 mg/kg were selected for RNA-seq in this study. RNA quality and quantity control results showed that both the Q30 value and the percentage of the total mapped reads were above 90%, and the preprocessed results met the requirements for in silico sequencing [24]. The results of RNA-seq showed that when the HMGCR inhibitors were added at a dose of 300 mg/kg, the number of differentially expressed genes in the liver of the laying hens increased by about 2.6 times compared to 150 mg/kg, which may have been due to the enhanced dose effect of the HMGCR inhibitors [20]. 

Elkin et al. reported that adding 300 mg/kg of HMGCR inhibitors to feed resulted in an upregulation of HMGCR activity in the liver of laying hens [20]. In this study, the genes related to cholesterol synthesis were significantly upregulated in the liver of laying hens after the addition of two different doses of HMGCR inhibitors, which is similar to Elkin et al.’s findings. The reason may be that the total amount of cholesterol synthesised by the body decreased after the synthesis of MVA was inhibited by HMGCR inhibitors [25]. To meet the daily demand of cholesterol, the body initiates a negative feedback regulatory response, which leads to the upregulation of all genes related to cholesterol synthesis [26].

The liver is an important site for the synthesis of apolipoproteins, which play a very important role in the transport of lipids and cholesterol [27,28]. In this study, HMGCR inhibitors significantly downregulated the mRNA expression levels of *APOC3* in the liver. The *APOC3* gene is involved in the synthesis of VLDL [29], which is mainly responsible for the transport of cholesterol into the yolk [30]. The decreased mRNA expression of the *APOC3* gene in the liver, as well as the decreased content of VLDL-C in the serum, are also important causes of the decreased cholesterol content in the yolk. 

Bile acids are synthesised by the liver and are an important component of cholesterol metabolism [31,32]. Regulating the secretion of bile acid and inhibiting the reabsorption of bile acid are important means to regulate cholesterol metabolism in medicine [33]. *CYP7A1* is a key gene that regulates bile acid synthesis, and *CYP8B1* is also involved in bile acid synthesis [34]. In this study, both *CYP7A1* and *CYP8B1* genes were significantly upregulated in the liver after the addition of 300 mg/kg of the HMGCR inhibitor. The reason may be that after the reduction in cholesterol synthesis in the liver, the synthesis of bile acids is also inhibited, the supply of bile acids is insufficient, and the body initiates the negative feedback regulation effect, which leads to the synthesis and expression of the bile-acid-related gene being upregulated [35].

Reverse cholesterol transport can transport excess cholesterol from tissues to the liver for reuse [36]. The reverse cholesterol transporter (*CETP*) is an important gene that mediates this process [36,37]. In this study, *CETP* gene expression in the liver of laying hens was significantly downregulated after the addition of HMGCR inhibitors. This may have been due to the phenomenon of insufficient cholesterol supply to tissues after the amount of cholesterol synthesis in the liver decreased, and the body initiated a negative feedback regulation, which thereby reduced the reverse transport of cholesterol in peripheral tissues [26]. To test this speculation, this study selected the pectoral muscle as a representative peripheral tissue and measured its cholesterol content. The results showed that the TC content in the thoracic muscle decreased with the addition of the HMGCR inhibitor, which was consistent with our speculation.

Yolk is formed by the development of oocytes in the ovary [11], and although there are three cholesterol-carrying lipoproteins in the serum, only cholesterol carried by VLDL can be transported into oocytes through the stromal membrane of ovarian granulosa cells [11,12]. In this process, VLDL binds to the VLDLR on the ovary and transports cholesterol into the developing oocyte in a receptor-mediated manner [11,12]. In this study, the mRNA expression of *VLDLR* in the ovary was examined, and the results showed that the mRNA expression of *VLDLR* was significantly upregulated with the addition of the HMGCR inhibitor. The reason may be that the total amount of VLDL in the serum was reduced, which led to a lower amount of VLDL entering the ovary, and the body initiated the negative feedback regulatory effect, which resulted in a significant upregulation of the *VLDLR* mRNA [26,35].

Combining the results of the RNA-seq and the determination of the cholesterol content in individual tissues, a pathway diagram was drawn of the biological mechanism by which the HMGCR inhibitors reduced the yolk cholesterol content (Figure 3). Compared with the mouse cholesterol metabolism pathway [38], some key genes were not differentially expressed in this study. It may be that the differences between laying chicken and mouse species are larger or that HMGCR inhibitors have no effect on the expression of these genes, which are specific reasons that require further investigation.

## 5. Conclusions

In summary, HMGCR inhibitors effectively inhibit hepatic cholesterol synthesis and reduce cholesterol content in the liver, serum, yolk, and breast muscle of laying hens, with the lowest effective dose of HMGCR inhibitors being 150 mg/kg in laying hens. In addition, HMGCR inhibitors reduce yolk cholesterol mainly by regulating the pathways involved in hepatic cholesterol and bile acid synthesis, cholesterol transport, and the reverse transport of cholesterol from peripheral tissues.

## Figures and Tables

**Figure 1 animals-13-01868-f001:**
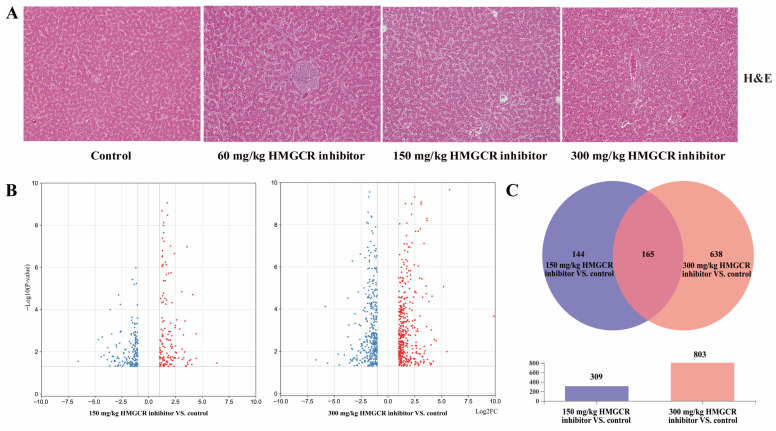
Effects of HMGCR inhibitors on liver tissue of laying hens and genes differentially expressed in the RNA-seq (Red indicates up regulation, blue indicates down regulation). (**A**) Pathological sections of the liver (×200); (**B**) volcanic diagram of differentially expressed genes (red means up, blue means down); (**C**) Wayne diagram of differentially expressed genes coexpression.

**Figure 2 animals-13-01868-f002:**
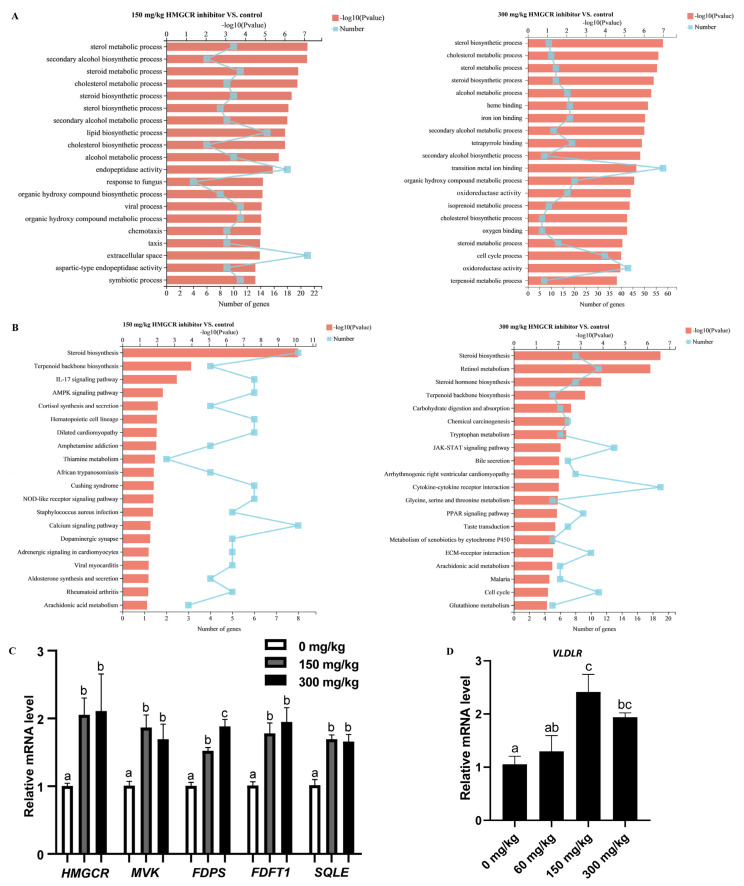
Analysis of RNA-seq and q-PCR. (**A**) GO enrichment analysis of differentially expressed genes; (**B**) KEGG pathway enrichment analysis of differentially expressed genes; (**C**) verifying the accuracy of RNA seq through q-PCR (*n* = 6); (**D**) expression of *VLDLR* gene in ovary (*n* = 6). ^a,b,c^ Columns with different owercase letters indicated significant differences between the compared groups (*p* < 0.05).

**Figure 3 animals-13-01868-f003:**
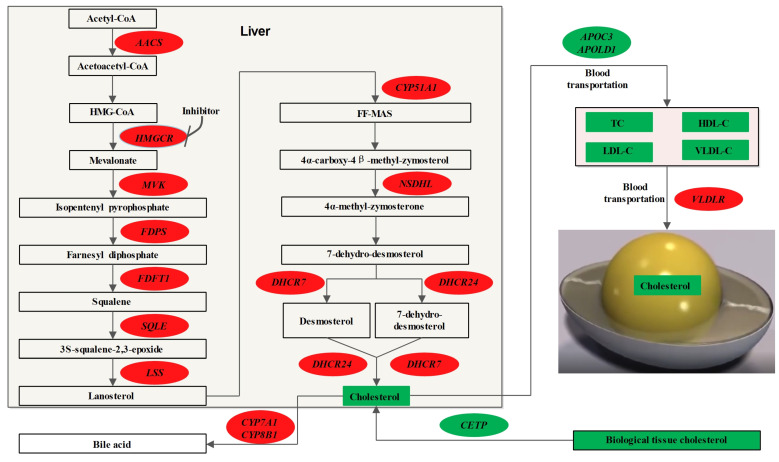
Effects of HMGCR inhibitor on cholesterol metabolism in laying hens. Oval circle represents gene and square box represents metabolite; red represents upregulation and green represents downregulation.

**Table 1 animals-13-01868-t001:** Statistical summary of the liver RNA-seq datasets.

Sample ^1^	Raw Reads Number	Q30 Value ^2^	Clean Reads Number	Total Mapped Reads Percentage
Control-1	59,376,004	94.26	57,428,878	92.44
Control-2	45,215,482	94.29	43,325,868	92.07
Control-3	46,066,730	93.96	44,298,418	92.52
Control-4	56,379,500	94.35	54,072,414	91.23
Control-5	53,838,578	94.09	52,314,524	92.71
Control-6	47,609,604	94.02	45,817,648	92.18
HI-150-1	49,822,070	94.07	48,155,306	92.12
HI-150-2	57,097,152	94.30	55,173,228	92.79
HI-150-3	51,455,312	94.39	49,677,616	91.70
HI-150-4	62,788,508	94.12	60,970,162	92.02
HI-150-5	52,293,100	94.56	51,003,728	92.22
HI-150-6	48,378,732	94.51	46,949,796	92.47
HI-300-1	48,598,154	94.32	47,124,918	91.84
HI-300-2	52,669,602	94.06	51,208,282	92.04
HI-300-3	47,767,574	93.98	46,138,424	92.11
HI-300-4	51,883,822	93.97	50,588,624	92.57
HI-300-5	47,296,884	93.55	45,502,646	92.14
HI-300-6	51,617,102	94.05	50,053,864	91.95

^1^ Control, HI-150, and HI-300 means the liver samples from the diets supplemented with HMGCR inhibitor at doses of 0, 150, and 300 mg/kg, respectively. ^2^ Q30 value means the sequencing quality values that correspond to 0.1% chance of error.

**Table 2 animals-13-01868-t002:** Changes in egg quality ^1^.

HMGCR Inhibitor Dosage		Control	60 mg/kg	150 mg/kg	300 mg/kg	SEM
Week 2	Egg weight (g)	54.30 ± 1.69 ^a^	49.97 ± 1.85 ^b^	47.59 ± 3.32 ^b^	48.40 ± 2.63 ^b^	3.46
	Egg yolk weight (g)	13.72 ± 0.76 ^a^	12.78 ± 0.44 ^b^	12.14 ± 0.85 ^bc^	11.89 ± 0.84 ^c^	0.99
	Proportion of egg yolk (%)	25.27 ± 0.85	25.58 ± 0.79	25.53 ± 1.25	24.58 ± 1.57	1.14
	Haugh unit	88.03 ± 11.84	88.58 ± 5.16	86.64 ± 11.43	85.21 ± 5.18	8.52
	Albumen height	7.95 ± 1.81	7.89 ± 1.21	7.60 ± 2.13	7.39 ± 0.90	1.50
	Yolk colour	12.62 ± 0.74	12.75 ± 0.89	13.00 ± 1.07	13.37 ± 0.91	0.90
Week 4	Egg weight (g)	54.46 ± 2.22 ^a^	50.15 ± 2.10 ^b^	46.43 ± 2.18 ^c^	43.41 ± 3.13 ^d^	4.73
	Egg yolk weight (g)	13.48 ± 0.69 ^a^	12.23 ± 0.84 ^b^	10.46 ± 0.80 ^c^	9.60 ± 0.79 ^d^	1.68
	Proportion of egg yolk (%)	24.77 ± 0.83 ^a^	24.37 ± 1.23 ^a^	22.57 ± 1.84 ^b^	22.20 ± 2.19 ^b^	1.98
	Haugh unit	84.00 ± 9.25	86.09 ± 12.98	82.05 ± 20.23	82.18 ± 12.60	13.51
	Albumen height	7.16 ± 1.86	7.78 ± 2.74	6.98 ± 3.34	6.79 ± 1.80	2.39
	Yolk colour	13.63 ± 0.92 ^a^	13.63 ± 0.92 ^a^	13.50 ± 1.20 ^a^	11.75 ± 0.89 ^b^	1.22

^1^ Values are expressed as means ± SD (*n* = 8), and different superscripts in different columns of the same row indicate significant differences (*p* < 0.05).

**Table 3 animals-13-01868-t003:** Changes in TC and TG content in tissues ^1^.

HMGCR Inhibitor Dosage		Control	60 mg/kg	150 mg/kg	300 mg/kg	SEM
Week 2	Serum TC (mmol/L)	3.65 ± 0.37 ^a^	2.96 ± 0.27 ^b^	2.50 ± 0.12 ^c^	1.85 ± 0.21 ^d^	0.70
	Serum TG (mmol/L)	13.18 ± 1.45 ^a^	12.18 ± 1.15 ^a^	10.37 ± 0.98 ^b^	7.83 ± 1.55 ^c^	2.36
Week 4	Serum TC (mmol/L)	3.42 ± 0.27 ^a^	3.05 ± 0.19 ^b^	2.64 ± 0.22 ^c^	1.92 ± 0.30 ^d^	0.60
	Serum HDL-C (mmol/L)	0.78 ± 0.15 ^a^	0.73 ± 0.14 ^ab^	0.61 ± 0.09 ^bc^	0.57 ± 0.15 ^c^	0.15
	Serum LDL-C (mmol/L)	2.03 ± 0.75 ^a^	1.72 ± 0.79 ^a^	1.38 ± 0.59 ^ab^	0.97 ± 0.59 ^b^	0.75
	Serum VLDL-C (mmol/L)	0.83 ± 0.37 ^a^	0.72 ± 0.26 ^ab^	0.59 ± 0.24 ^ab^	0.52 ± 0.19 ^b^	0.28
	Liver TC (mg/g)	2.68 ± 0.39 ^a^	2.53 ± 0.43 ^a^	2.29 ± 0.53 ^ab^	1.91 ± 0.29 ^b^	0.49
	Pectoralis TC (mg/g)	0.52 ± 0.02 ^a^	0.50 ± 0.03 ^b^	0.46 ± 0.02 ^c^	0.43 ± 0.02 ^d^	0.04
	Serum TG (mmol/L)	13.65 ± 0.74 ^a^	12.72 ± 0.84 ^a^	10.49 ± 2.30 ^b^	6.83 ± 0.90 ^c^	2.90

^1^ Values are expressed as means ± SD (*n* = 8), and different superscripts in different columns of the same row indicate significant differences (*p* < 0.05).

**Table 4 animals-13-01868-t004:** Changes in cholesterol content in eggs ^1^.

HMGCR Inhibitor Dosage		Control	60 mg/kg	150 mg/kg	300 mg/kg	SEM
Week 2	Yolk TC (mg/g)	13.99 ± 0.49 ^a^	13.66 ± 0.56 ^a^	12.88 ± 0.40 ^b^	12.24 ± 0.53 ^c^	0.83
	Eggs TC (mg)	191.85 ± 9.19 ^a^	174.62 ± 10.78 ^b^	156.33 ± 12.89 ^c^	145.41 ± 10.51 ^c^	20.45
	Proportion of eggs cholesterol (mg/g)	3.53 ± 0.14 ^a^	3.50 ± 0.22 ^a^	3.28 ± 0.12 ^b^	3.01 ± 0.20 ^c^	0.27
Week 4	Yolk TC (mg/g)	13.85 ± 0.74 ^a^	13.33 ± 1.11 ^ab^	12.34 ± 1.09 ^bc^	11.53 ± 1.06 ^c^	1.30
	Eggs TC (mg)	186.93 ± 15.43 ^a^	162.62 ± 13.38 ^b^	129.44 ± 18.50 ^c^	110.41 ± 10.66 ^d^	32.59
	Proportion of eggs cholesterol (mg/g)	3.43 ± 0.25 ^a^	3.25 ± 0.32 ^a^	2.79 ± 0.43 ^b^	2.55 ± 0.24 ^b^	0.46

^1^ Values are expressed as means ± SD (*n* = 8), and different superscripts in different columns of the same row indicate significant differences (*p* < 0.05).

**Table 5 animals-13-01868-t005:** Differential genes in enriched pathways in the liver of laying hens.

Gene ID	Gene Symbol	Log2 (Fold Change) (150) ^1^	Log2 (Fold Change) (300) ^2^	Gene Description
Liver cholesterol synthesis				
416811	*AACS*	1.269	2.034	Acetoacetyl-CoA synthetase
395145	*HMGCR*	1.396	1.400	3-Hydroxy-3-methylglutaryl-CoA reductase
768555	*MVK*	1.294	1.328	Mevalonate kinase
425061	*FDPS*	1.861	1.839	Farnesyl diphosphate synthase
422423	*MSMO1*	1.792	1.642	Methylsterol monooxygenase 1
422038	*FDFT1*	1.520	1.532	Farnesyl-diphosphate farnesyltransferase 1
420335	*SQLE*	2.150	2.098	Squalene epoxidase
424037	*LSS*	1.351	1.287	Lanosterol synthase
420548	*CYP51A1*	1.751	1.678	Cytochrome P450 family 51
422302	*NSDHL*	1.649	1.482	NAD(P) dependent steroid dehydrogenase-like
422982	*DHCR7*	1.553	1.673	7-Dehydrocholesterol reductase
424661	*DHCR24*	1.430	1.497	24-Dehydrocholesterol reductase
Cholesterol transport				
100859721	*APOC3*	/	−1.211	Apolipoprotein C3
769889	*APOLD1*	−1.479	−1.228	Apolipoprotein L domain containing 1
Bile acid synthesis				
414834	*CYP7A1*	/	1.502	Cytochrome P450 family 7
425055	*CYP8B1*	/	2.384	Cytochrome P450 family 8
Peripheral tissue reverse cholesterol transport				
415645	*CETP*	−2.639	−3.321	Cholesteryl ester transfer protein

^1^ “150” means that the group with 150 mg/kg HMGCR inhibitor was compared with the control group. ^2^ “300” means that the group with 300 mg/kg HMGCR inhibitor was compared with the control group.

## Data Availability

Raw data are held by the author and may be available upon request.

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
