# Peer review of "Effects of 3-Hydroxy-3-methylglutaryl-CoA Reductase Inhibitors on Cholesterol Metabolism in Laying Hens"

_animals, 2023, doi:10.3390/ani13111868_

Round 1

Reviewer 1 Report

This study investigated effects of dietary atorvastatin on cholesterol metabolism in laying hens. The experimental strategies are straightforward. Although this study shows a large amount of data, there are some concerns and issues to address.

Major comments.

1. The aim of this study is unclear. "No report" (Line 64-66) is not good as the aim of the study. In addition, an industrial significance is not shown in this manuscript. Do the authors believe that consumers accept eggs from laying hens fed with atorvastatin?

2. The authors do not provide a precise description about lipoprotein in poultry. Chylomicron does not exist in chickens. A specific VLDL that differ from normal VLDL is synthesized in laying hens. The authors should revise the related description with appropriate references.

3. Statistical method is inappropriate. t-test cannot be used in multiple comparison. For example, Tukey-Kramer method and Dunnett' test should be used.

4. The Discussion is not well organized. There are numerous issues, and the main points are as follows. 

Line 274-292 is not discussion. These sentences should be written in the Results.

Ref 26 is not appropriate (Line 300). Cell, 89, 331-340, 1997 seems to be more appropriate.

Although the authors discuss about several genes based on the results of RNA-seq, they should verify the expression of the genes by real-time PCR  like Fig 2C and D.

The description about APOB (Line 304-309) seems to be unnecessary. How do the authors think APOB expression is related to those of APOC3 and APOLD1? 

Why do the authors think total cholesterol contents in the pectoral muscle prove their speculation?

Minor comments.

Line 98 and 157: "protein" height appears to be error. "Albumen" height is correct.

Line 239: Delete "The".

Line 241: Spell out "MVA".

Line 252-253 and 348-350: It is difficult to understand what the authors mean.

Line 341: "VLDLR" appears to be error. "VLDL" is correct.

Order of figures and tables: Tables are not presented in order of the results in this manuscript. Table 2-4 should be presented prior to Table 1. The results in Fig 1A (liver section) and Fig 1BC (RNA-seq) are not related and should be presented separately.

Suppl Table 2: Amplification efficiency should be shown, because the authors analyzed gene expression by delta-delta Ct method.

Author Response

请参阅附件。

Reviewer 2 Report

Dear Author,

Greetings of the day,

I have some comments about this manuscript that have been shown below:

Line 2: Normally the abbreviation will not allow in the title!

Line 18: Authors have to check this number (180) it should be 192! check Line 78 as well.

Line 78: Check these two numbers 192 and 180!!!

Line 86: What is the reason to choose 60 mg! why not 75 so as to half of 150 and 300 double of 150!

Line 87: In which age? The age of layers?

Line 119: Why authors have done gene expression just for 150 and 300! where is 60!

Line 147: Normally SPSS use for analyzing social sciences! it would be better if analyzed by SAS.

Line 160: Table 2, I suggest authors to remove the plus or minus standard deviations for every treatment and use one column that has standard error of the means (SEM). This will be much easier for readers to understand.

Line 184: Authors have to add (with in column)

Line 229: Figure 2-c, It is better if authors write fold change in Y axis!

Line 232: Authors should mention VLDLR in the figure so as to be clear without return back to title.

Line 239: delete ‘’The’’

Line 346: Authors should delete these are figures in discussion part! it will be better if explain it in text.

Regards,

Reviewer 3 Report

This manuscript states that administration of HMGCR inhibitors to chickens can reduce the cholesterol content in eggs.

There are no points of concern in the central part of this manuscript.Moreover, the cited and their discussion are presented in good style. 

I have the following questions and suggestions for the authors.

The authors argue that the lack of significant reduction in egg cholesterol in the 60 mg/day statin group may be due to the lack of reduction in VLDLC.

Why not look further and discuss cholesterol homeostasis?

e.g.) Ingested cholesterol + biosynthesized cholesterol = const.

Also, are there any investigations or previous studies that show a further decrease with concomitant use of cholestyramine, etc.?

no problem

Round 2

Reviewer 1 Report

Although the authors respond to the reviewer 1's comments, a few minor changes are still required.

Line 100 and 159: "Albumen" should start with lowercase.

Line 153: A hyphen (-) is required between "Tukey" and "Kramer".

Line 240: Spell out "VLDLR".

Line 297-309: This paragraph seems to discuss the relationship between downregulation of APOC3 expression and the decreased cholesterol content in the yolk. As commented previously, if there is no relationship between APOC3 and APOB, the sentences about APOB (line 300-305) are unnecessary in this paragraph. In addition, APOLD1 also seems to be unnecessary. If the authors discuss discrepancies between the present study and previous studies in APOB expression, it would be better to discuss them in another paragraph.

Line 327-329: It would be better to add the description that the authors analyzed the TC content in the pectoral muscle as the representative peripheral tissue.

Line 337: Does VLDLR exist in the serum of chickens? Why does the reduction of serum VLDLR lead to the decrease of the amount of VLDL entering the ovary?

Legend of Figure 2: Add the description of abbreviations of genes.
